# D4FT: A Deep Learning Approach to Kohn-Sham Density Functional Theory

**Tianbo Li**[*][a, e], **Min Lin**[*][a, f], **Zheyuan Hu**[a, b], **Kunhao Zheng**[a], **Giovanni Vignale**[c],
**Kenji Kawaguchi**[b], **A. H. Castro Neto**[c, d], **Kostya S. Novoselov**[c, d], **Shuicheng Yan**[a]
[a]SEA AI Lab, [b]School of Computing, National University of Singapore,
[c]Institute for Functional Intelligent Materials, National University of Singapore,
[d]Centre for Advanced 2D Materials, National University of Singapore;
{[e]litb, [f]linmin}@sea.com

## Abstract

Kohn-Sham Density Functional Theory (KS-DFT) has been traditionally solved by the Self-Consistent Field (SCF) method. Behind the SCF loop is the physics intuition of solving a system of non-interactive single-electron wave functions under an effective potential. In this work, we propose a deep learning approach to KS-DFT. First, in contrast to the conventional SCF loop, we propose directly minimizing the total energy by reparameterizing the orthogonal constraint as a feed-forward computation. We prove that such an approach has the same expressivity as the SCF method yet reduces the computational complexity from $\mathcal{O}(N^4)$ to $\mathcal{O}(N^3)$. Second, the numerical integration, which involves a summation over the quadrature grids, can be amortized to the optimization steps. At each step, stochastic gradient descent (SGD) is performed with a sampled minibatch of the grids. Extensive experiments are carried out to demonstrate the advantage of our approach in terms of efficiency and stability. In addition, we show that our approach enables us to explore more complex neural-based wave functions.

## 1 Introduction

Density functional theory (DFT) is the most successful quantum-mechanical method, which is widely used in chemistry and physics for predicting electron-related properties of matters (Szabo & Ostlund, 2012; Levine et al., 2009; Koch & Holthausen, 2015). As scientists are exploring more complex molecules and materials, DFT methods are often limited in scale or accuracy due to their computation complexity. On the other hand, Deep Learning (DL) has achieved great success in function approximations (Hornik et al., 1989), optimization algorithms (Kingma & Ba, 2014), and systems (Bradbury et al., 2018) in the past decade. Many aspects of deep learning can be harnessed to improve DFT. Of them, data-driven function fitting is the most straightforward and often the first to be considered. It has been shown that models learned from a sufficient amount of data generalize greatly to unseen data, given that the models have the right inductive bias. The Hohenberg-Kohn theorem proves that the ground state energy is a functional of electron density (Hohenberg & Kohn, 1964a), but this functional is not available analytically. This is where data-driven learning can be helpful for DFT. The strong function approximation capability of deep learning gives hope to learning such functionals in a data-driven manner. There have already been initial successes in learning the exchange-correlation functional (Chen et al., 2020a;b; Dick & Fernandez-Serra, 2020). Furthermore, deep learning has shifted the mindsets of researchers and engineers towards differentiable programming. Implementing the derivative of a function has no extra cost if the primal function is implemented with deep learning frameworks. Derivation of functions frequently appears in DFT, e.g., estimating the kinetic energy of a wave function; calculating generalized gradient approximation (GGA) exchange-correlation functional, etc. Using modern automatic differentiation (AD) techniques ease the implementation greatly (Abbott et al., 2021).

Despite the numerous efforts that apply deep learning to DFT, there is still a vast space for exploration. For example, the most popular Kohn-Sham DFT (KS-DFT) (Kohn & Sham, 1965) utilizes

---

[*] Equal Contribution. Our code will be available on https://github.com/sail-sg/d4ft.

the self-consistency field (SCF) method for solving the parameters. At each SCF step, it solves a closed-form eigendecomposition problem, which finally leads to energy minimization. However, this method suffers from many drawbacks. Many computational chemists and material scientists criticize that optimizing via SCF is time-consuming for large molecules or solid cells, and that the convergence of SCF is not always guaranteed. Furthermore, DFT methods often utilize the linear combination of basis functions as the ansatz of wave functions, which may not have satisfactory expressiveness to approximate realistic quantum systems.

To address the problems of SCF, we propose a deep learning approach for solving KS-DFT. Our approach differs from SCF in the following aspects. First, the eigendecomposition steps in SCF come from the orthogonal constraints on the wave functions; we show in this work that the original objective function for KS-DFT can be converted into an unconstrained equivalent by reparameterizing the orthogonal constraints as part of the objective function. Second, we further explore amortizing the integral in the objective function over the optimization steps, i.e., using stochastic gradient descent (SGD), which is well-motivated both empirically and theoretically for large-scale machine learning (Bottou et al., 2018). We demonstrate the equivalence between our approach and the conventional SCF both empirically and theoretically. Our approach reduces the computational complexity from $\mathcal{O}(N^4)$ to $\mathcal{O}(N^3)$, which significantly improves the efficiency and scalability of KS-DFT. Third, gradient-based optimization treats all parameters equally. We show that it is possible to optimize more complex neural-based wave functions instead of optimizing only the coefficients. In this paper, we instantiate this idea with local scaling transformation as an example showing how to construct neural-based wave functions for DFT.

## 2 DFT Preliminaries

Density functional theory (DFT) is among the most successful quantum-mechanical simulation methods for computing electronic structure and all electron-related properties. DFT defines the ground state energy as a functional of the electron density $\rho : \mathbb{R}^3 \to \mathbb{R}$:

$$E_{\text{gs}} = E[\rho]. \tag{1}$$

The Hohenberg-Kohn theorem (Hohenberg & Kohn, 1964b) guarantees that such functionals $E$ exists and the ground state energy can be determined uniquely by the electron density. However, the exact definition of such functional has been a puzzling obstacle for physicists and chemists. Some approximations, including the famous Thomas-Fermi method and Kohn-Sham method, have been proposed and have later become the most important ab-initio calculation methods.

**The Objective Function** One of the difficulties in finding the functional of electron density is the lack of an accurate functional of the kinetic energy. The Kohn-Sham method resolves this issue by introducing an orthogonal set of single-particle wave functions $\{\psi_i\}$ and rewriting the energy as a functional of these wave functions. The energy functional connects back to the Schrödinger equation. Without compromising the understanding of this paper, we leave the detailed derivation from Schrödinger equation and the motivation of the orthogonality constraint in Appendix B.1. As far as this paper is concerned, we focus on the objective function of KS-DFT, defined as,

$$E_{\text{gs}} = \min_{\{\psi_i^\sigma\}} E[\{\psi_i^\sigma\}] \tag{2}$$

$$= \min_{\{\psi_i^\sigma\}} E_{\text{Kin}}[\{\psi_i^\sigma\}] + E_{\text{Ext}}[\{\psi_i^\sigma\}] + E_{\text{H}}[\{\psi_i^\sigma\}] + E_{\text{XC}}[\{\psi_i^\sigma\}] \tag{3}$$

$$\text{s.t.} \quad \langle \psi_i^\sigma | \psi_j^\sigma \rangle = \delta_{ij} \tag{4}$$

where $\psi_i^\sigma$ is a wave function mapping $\mathbb{R}^3 \to \mathbb{C}$, and $\psi_i^{\sigma*}$ denotes its complex conjugate. For simplicity, we use the bra-ket notation for $\langle \psi_i^\sigma | \psi_j^\sigma \rangle = \int \psi_i^{\sigma*}(\boldsymbol{r}) \psi_j^\sigma(\boldsymbol{r}) d\boldsymbol{r}$. $\delta_{ij}$ is the Kronecker delta function. The superscript $\sigma \in \{\alpha, \beta\}$ denotes the spin.[1] $E_{\text{Kin}}$, $E_{\text{Ext}}$, $E_{\text{H}}$, $E_{\text{XC}}$ are the kinetic, external potential (nuclear attraction), Hartree (Coulomb repulsion between electrons) and

---

[1] We omit the spin notation $\sigma$ in the following sections for simplification reasons.

exchange-correlation energies respectively, defined by

$$E_{\text{Kin}}[\{\psi_i^\sigma\}] = -\frac{1}{2} \sum_i^N \sum_\sigma \int \psi_i^{\sigma *}(\boldsymbol{r}) \left(\nabla^2 \psi_i^\sigma(\boldsymbol{r})\right) d\boldsymbol{r}, \tag{5}$$

$$E_{\text{Ext}}[\{\psi_i^\sigma\}] = \sum_i^N \sum_\sigma \int v_{\text{ext}}(\boldsymbol{r})|\psi_i^\sigma(\boldsymbol{r})|^2 d\boldsymbol{r}, \tag{6}$$

$$E_{\text{H}}[\{\psi_i^\sigma\}] = \frac{1}{2} \int \int \frac{\left(\sum_i^N \sum_\sigma |\psi_i^\sigma(\boldsymbol{r})|^2\right)\left(\sum_j^N \sum_\sigma |\psi_j^\sigma(\boldsymbol{r}')|^2\right)}{|\boldsymbol{r} - \boldsymbol{r}'|} d\boldsymbol{r} d\boldsymbol{r}', \tag{7}$$

$$E_{\text{XC}}[\{\psi_i^\sigma\}] = \sum_i^N \sum_\sigma \int \varepsilon_{\text{xc}}(\boldsymbol{r})|\psi_i^\sigma(\boldsymbol{r})|^2 d\boldsymbol{r}, \tag{8}$$

in which $v_{\text{ext}}$ is the external potential defined by the molecule's geometry. $\varepsilon_{\text{xc}}$ is the exchange-correlation energy density, which has different instantiations. The entire objective function is given analytically, except for the function $\psi_i^\sigma$ that we replace with parametric functions to be optimized. In the rest of this paper, we focus on the algorithms that optimize this objective while minimizing the discussion on its scientific background.

**Kohn-Sham Equation and SCF Method** The object function above can be solved by Euler–Lagrange method, which yields the canonical form of the well-known Kohn-Sham equation,

$$\left[-\frac{1}{2}\nabla^2 + v_{\text{ext}}(\boldsymbol{r}) + \int d^3\boldsymbol{r}' \frac{\sum_i^N |\psi_i(\boldsymbol{r}')|^2}{|\boldsymbol{r} - \boldsymbol{r}'|} + v_{\text{xc}}(\boldsymbol{r})\right] \psi_i = \varepsilon_i \psi_i \tag{9}$$

where $N$ is the total number of electrons. $v_{\text{ext}}$ and $v_{\text{xc}}$ are the external and exchange-correlation potentials, respectively. This equation is usually solved in an iterative manner called Self-Consistent Field (SCF). This method starts with an initial guess of the orthogonal set of single electron wave functions, which are then used for constructing the Hamiltonian operator. The new wave functions and corresponding electron density can be obtained by solving the eigenvalue equation. This process is repeated until the electron density of the system converges. The derivation of the Kohn-Sham equation and the SCF algorithm is presented in Appendix B.2.

**The LCAO Method** In the general case, we can use any parametric approximator for $\psi_i$ to transform the optimization problem from the function space to the parameter space. In quantum chemistry, they are usually represented by the *linear combination of atomic orbitals* (LCAO):

$$\psi_i(\boldsymbol{r}) = \sum_j^B c_{ij}\phi_j(\boldsymbol{r}), \tag{10}$$

where $\phi_j$ are atomic orbitals (or basis functions more generally), which are usually pre-determined analytical functions, e.g., truncated series of spherical harmonics or plane waves. We denote the number of the basis functions as $B$. The single-particle wave functions are linear combinations of these basis functions with $c_{ij}$ as the only optimizable parameters. We introduce vectorized notations $\boldsymbol{\Psi} := (\psi_1, \psi_2, \cdots, \psi_N)^\top$, $\boldsymbol{\Phi} := (\phi_1, \phi_2, \cdots, \phi_N)^\top$ and $\boldsymbol{C} := [c_{ij}]$, so the LCAO wave functions can be written as,

$$\boldsymbol{\Psi} = \boldsymbol{C}\boldsymbol{\Phi}. \tag{11}$$

Classical atomic obitals include Slater-type orbitals, Pople basis sets (Ditchfield et al., 1971), correlation-consistent basis sets (Dunning Jr, 1989), etc.

## 3  A DEEP LEARNING APPROACH TO KS-DFT

In this section, we propose a deep learning approach to solving KS-DFT. Our method can be described by three keywords:

- *deep learning*: our method is deep learning native; it is implemented in a widely-used deep learning framework JAX;
- *differentiable*: all the functions are differentiable, and thus the energy can be optimized via purely gradient-based methods;

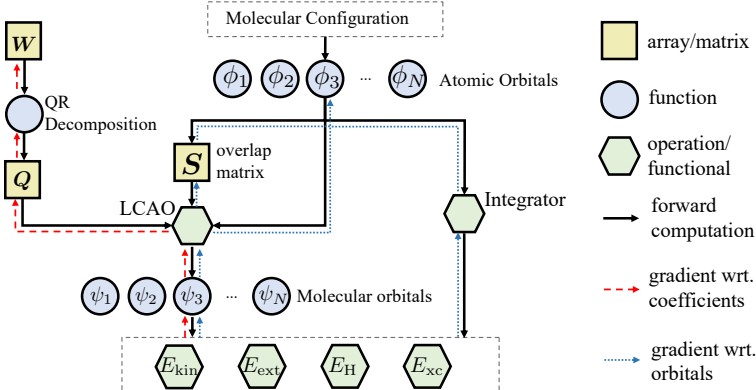

Figure 1: An illustration on the computational graph of D4FT. In conventional DFT methods using fixed basis sets, the gradients w.r.t. the basis functions (blue dotted line) are unnecessary.

- *direct optimization*: due to the differentiability, our method does not require self-consistent field iterations, but the convergence result is also self-consistent.

We refer to our method as **D4FT** that represents the above 3 keywords and DFT. Fig. 1 shows an overarching framework of our method. We elaborate on each component in the following parts.

## 3.1 REPARAMETERIZE THE ORTHONORMAL CONSTRAINTS

Constraints can be handled with various optimization methods, e.g., the Lagrangian multiplier method or the penalty method. In deep learning, it is preferred to reparameterize the constraints into the computation graph. For example, when we want the normalization constraint on a vector $x$, we can reparameterize it to $y/\|y\|$ such that it converts to solving $y$ without constraints. Then this constraint-free optimization can benefit from the existing differentiable frameworks and various gradient descent optimizers. The wave functions in Kohn-Sham DFT have to satisfy the constraint $\langle\psi_i|\psi_j\rangle = \delta_{ij}$ as given in Eq. 4. Traditionally the constraint is handled by the Lagrangian multiplier method, which leads to the SCF method introduced in Sec. 2. To enable direct optimization, we propose the following reparameterization of the constraint.

Using LCAO wave functions, the constraint translates to $\int \sum_k c_{ik}\phi_k(\boldsymbol{r}) \sum_l c_{jl}\phi_l(\boldsymbol{r})d\boldsymbol{r} = \delta_{ij}$. Or in the matrix form

$$\boldsymbol{C}\boldsymbol{S}\boldsymbol{C}^\top = I. \tag{12}$$

$\boldsymbol{S}$ is called *the overlap matrix* of the basis functions, where the $ij$-th entry $S_{ij} = \langle\phi_i|\phi_j\rangle$. The literature on whitening transformation (Kessy et al., 2018) offers many ways to construct $\boldsymbol{C}$ to satisfy Eq. 12 based on different matrix factorization of $\boldsymbol{S}$:

$$\boldsymbol{C} = \begin{cases} \boldsymbol{Q}\boldsymbol{\Lambda}^{-1/2}\boldsymbol{U} & \text{PCA whitening, with } \boldsymbol{U}^\top\boldsymbol{\Lambda}\boldsymbol{U} = \boldsymbol{S} \\ \boldsymbol{Q}\boldsymbol{L}^\top & \text{Cholesky whitening, with } \boldsymbol{L}\boldsymbol{L}^\top = \boldsymbol{S}^{-1} \\ \boldsymbol{Q}\boldsymbol{S}^{-1/2} & \text{ZCA whitening} \end{cases} \tag{13}$$

Taking PCA whitening as an example. Since $\boldsymbol{S}$ can be precomputed from the overlap of basis functions, $\boldsymbol{\Lambda}^{-1/2}\boldsymbol{U}$ is fixed. The $\boldsymbol{Q}$ matrix can be any orthonormal matrix; for deep learning, we can parameterize $\boldsymbol{Q}$ in a differentiable way using QR decomposition of an unconstrained matrix $\boldsymbol{W}$:

$$\boldsymbol{Q}, \boldsymbol{R} = \mathrm{QR}(\boldsymbol{W}). \tag{14}$$

Besides QR decomposition, there are several other differentiable methods to construct the orthonormal matrix $\boldsymbol{Q}$, e.g., Householder transformation (Mathiasen et al., 2020), and exponential map (Lezcano-Casado & Martınez-Rubio, 2019). Finally, the wave functions can be written as:

$$\boldsymbol{\Psi}_{\boldsymbol{W}} = \boldsymbol{Q}\boldsymbol{D}\boldsymbol{\Phi}. \tag{15}$$

In this way, the wave functions $\boldsymbol{\Psi}_{\boldsymbol{W}}$ are always orthogonal given arbitrary $\boldsymbol{W}$. The searching over the orthogonal function space is transformed into optimizing over the parameter space that $\boldsymbol{W}$ resides. Moreover, this parameterization covers all possible sets of orthonormal wave functions in the space spanned by the basis functions. These statements are formalized in the following proposition, and the proof is presented in Appendix C.1.

**Proposition 3.1.** *Define the original orthogonal function space $\mathcal{F}$ and the transformed search space $\mathcal{F}_{\boldsymbol{W}}$ by $\mathcal{F} = \{\boldsymbol{\Psi} = (\psi_1, \psi_2, \cdots, \psi_N)^\top : \boldsymbol{\Psi} = \boldsymbol{C}\boldsymbol{\Phi}, \boldsymbol{C} \in \mathbb{R}^{N \times N}, \langle\psi_i|\psi_j\rangle = \delta_{ij}\}$ and $\mathcal{F}_{\boldsymbol{W}} = \{\boldsymbol{\Psi}_{\boldsymbol{W}} = (\psi_1^{\boldsymbol{W}}, \psi_2^{\boldsymbol{W}}, \cdots, \psi_N^{\boldsymbol{W}})^\top : \boldsymbol{\Psi}_{\boldsymbol{W}} = \boldsymbol{Q}\boldsymbol{D}\boldsymbol{\Phi}, (\boldsymbol{Q}, \boldsymbol{R}) = \mathrm{QR}(\boldsymbol{W}), \boldsymbol{W} \in \mathbb{R}^{N \times N}\}$. Then, they are equivalent to $\mathcal{F}_{\boldsymbol{W}} = \mathcal{F}$.*

## 3.2 STOCHASTIC GRADIENT

SGD is the modern workhorse for large-scale machine learning optimizations. It has been harnessed to achieve an unprecedented scale of training, which would have been impossible with full batch training. We elaborate in this section on how DFT could also benefit from SGD.

**Numerical Quadrature** The total energies defined in Equations 5-8 are integrals that involve the wave functions. Although the analytical solution to the integrals of commonly-used basis sets does exist, most DFT implementations adopt numerical quadrature integration, which approximate the value of a definite integral using a set of grids $\boldsymbol{g} = \{(\boldsymbol{x}_i, w_i)\}_{i=1}^n$ where $\boldsymbol{x}_i$ and $w_i$ are the coordinate and corresponding weights, respectively:

$$\int_a^b f(\boldsymbol{x})d\boldsymbol{x} \approx \sum_{\boldsymbol{x}_i, w_i \in \boldsymbol{g}} f(\boldsymbol{x}_i)w_i. \tag{16}$$

These grids and weights can be obtained via solving polynomial equations (Golub & Welsch, 1969; Abramowitz & Stegun, 1964). One key issue that hinders its application in large-scale systems is that the Hartree energy requires at least $O(n^2)$ calculation as it needs to compute the distance between every two grid points. Some large quantum systems will need 100k $\sim$ 10m grid points, which causes out-of-memory errors and hence is not feasible for most devices.

**Stochastic Gradient on Quadrature** Instead of evaluating the gradient of the total energy at all grid points in $\boldsymbol{g}$, we randomly sample a minibatch $\boldsymbol{g}' \subset \boldsymbol{g}$, where $\boldsymbol{g}'$ contains $m$ grid points, $m < n$, and evaluate the objective and its gradient on this minibatch. For example, for single integral energies such as kinetic energy, the gradient can be estimated by,

$$\widehat{\frac{\partial E_{kin}}{\partial \boldsymbol{W}}} = -\frac{1}{2}\frac{n}{m}\sum_{\boldsymbol{x}_i, w_i \in \boldsymbol{g}'} w_i \frac{\partial\left[\boldsymbol{\Psi}_{\boldsymbol{W}}^*(\boldsymbol{x}_i)(\nabla^2\boldsymbol{\Psi}_{\boldsymbol{W}})(\boldsymbol{x}_i)\right]}{\partial \boldsymbol{W}}. \tag{17}$$

The gradients of external and exchange-correlation energies can be defined accordingly. The gradient of Hartree energy, which is a double integral, can be defined as,

$$\widehat{\frac{\partial E_H}{\partial \boldsymbol{W}}} = \frac{2n(n-1)}{m(m-1)}\sum_{\boldsymbol{x}_i, w_i \in \boldsymbol{g}'}\sum_{\boldsymbol{x}_j, w_j \in \boldsymbol{g}'}\frac{w_i w_j\|\boldsymbol{\Psi}_{\boldsymbol{W}}(\boldsymbol{x}_j)\|^2}{\|\boldsymbol{x}_i - \boldsymbol{x}_j\|}\left[\sum_k \frac{\partial\psi_k(\boldsymbol{x}_i)}{\partial \boldsymbol{W}}\right]. \tag{18}$$

It can be proved that the expectation of the above stochastic gradient is equivalent to those of the full gradient. Note that the summation over the quadrature grids resembles the summation over all points in a dataset in deep learning. Therefore, in our implementation, we can directly rely on AD with minibatch to generate unbiased gradient estimates.

## 3.3 THEORETICAL PROPERTIES

**Asympototic Complexity** Here we analyze the computation complexity of D4FT in comparison with SCF. We use $N$ to denote the number of electrons, $B$ to denote the number of basis functions, $n$ for the number of grid points for quadrature integral, and $m$ for the minibatch size when the stochastic gradient is used. The major source of complexity comes from the hartree energy $E_H$, as it includes a double integral.

In our direct optimization approach, computing the hartree energy involves computing $\sum_i |\psi_i(\boldsymbol{r})|^2$ for all grid points, which takes $\mathcal{O}(nNB)$. After that $\mathcal{O}(n^2)$ computation is needed to compute and aggregate repulsions energies between all the electron pairs. Therefore, the total complexity is $\mathcal{O}(nNB) + \mathcal{O}(n^2)$. [2] In the SCF approach, it costs $\mathcal{O}(nNB) + \mathcal{O}(n^2N^2)$, a lot more expensive because it computes the Fock matrix instead of scalar energy.

---

[2] We assume the backward mode gradient computation has the same complexity as the forward computation.

Considering both $n$ and $B$ are approximately linear to $N$, the direct optimization approach has a more favorable $\mathcal{O}(N^3)$ compared to $\mathcal{O}(N^4)$ for SCF. However, since $n$ is often much bigger than $N$ and $B$ practically, the minibatch SGD is an indispensable ingredient for computation efficiency. At each iteration, minibatch reduces the factor of $n$ to a small constant $m$, making it $\mathcal{O}(mNB)+\mathcal{O}(m^2)$ for D4FT. A full breakdown of the complexity is available in Appendix D.

**Self-Consistency** The direct optimization method will converge at a self-consistent point. We demonstrate this in the following. We first give the definition of self-consistency as follows.

**Definition 3.1** (Self-consistency). *The wave functions $\boldsymbol{\Psi}$ is said to be self-consistent if the eigenfunctions of its corresponding Hamiltonian $\hat{H}(\boldsymbol{\Psi})$ is $\boldsymbol{\Psi}$ itself, i.e., there exists a real number $\varepsilon \in \mathbb{R}$, such that $\hat{H}(\boldsymbol{\Psi})|\boldsymbol{\Psi}\rangle = \varepsilon|\boldsymbol{\Psi}\rangle$.*

The next proposition states the equivalence between SCF and direct optimization. It states that the convergence point of D4FT is self-consistent. The proof is shown in the Appendix C.2.

**Proposition 3.2** (Equivalence between SCF and D4FT for KS-DFT). *Let $\boldsymbol{\Psi}^\dagger$ be a local optimimal of the ground state energy $\mathbb{E}_{gs}$ defined in Eq. 3, such that $\frac{\partial E_{gs}}{\partial \boldsymbol{\Psi}^\dagger} = \mathbf{0}$, then $\boldsymbol{\Psi}^\dagger$ is self-consistent.*

## 4 NEURAL BASIS WITH LOCAL SCALING TRANSFORMATION

In previous sections, the basis functions are considered given and fixed. Now we demonstrate and discuss the possibility of a learnable basis set that can be jointly optimized in D4FT. We hope that this extension could serve as a stepping stone toward neural-based wave function approximators.

As discussed in Sec. 2, LCAO wave functions are used in the existing DFT calculations. This restricts the wave functions to be in the subspace spanned by the basis functions, and it would then require a large number of basis functions in order to make a strong approximator. From the deep learning viewpoint, it is more efficient to increase the depth of the computation. The obstacle to using deep basis functions is that the overlap matrix $\boldsymbol{S}$ will change with the basis, requiring us to do expensive recomputation of whitening matrix $\boldsymbol{D}$ at each iteration. To remedy this problem, we introduce basis functions $\varphi_i$ in the following form

$$\varphi_i(\boldsymbol{r}) := \big| \det J_f(\boldsymbol{r})\big|^{\frac{1}{2}} \phi_i(f(\boldsymbol{r})) \tag{19}$$

where $f : \mathbb{R}^3 \to \mathbb{R}^3$ is a parametric bijective function, and $\det J_f(\boldsymbol{r})$ denotes its Jacobian determinant. It is verifiable by change of variable that

$$\langle \varphi_i | \varphi_j \rangle = \int |\det J_f(\boldsymbol{r})| \phi_i(f(\boldsymbol{r}))\phi_i(f(\boldsymbol{r}))d\boldsymbol{r} = \int \phi_i(\boldsymbol{u})\phi_j(\boldsymbol{u})d\boldsymbol{u} = \langle \phi_i | \phi_j \rangle.$$

Therefore, the overlap matrix $\boldsymbol{S}$ will be fixed and remain unchanged even if $f$ varies.

Within our framework, we can use a parameterized function $f_{\boldsymbol{\theta}}$ and optimize both $\boldsymbol{\theta}$ and $\boldsymbol{W}$ jointly with gradient descent. As a proof of concept, we design $f_{\boldsymbol{\theta}}$ as follows, which we term as *neural local scaling*:

$$f_{\boldsymbol{\theta}}(\boldsymbol{r}) := \lambda_{\boldsymbol{\theta}}(\boldsymbol{r})\boldsymbol{r}, \tag{20}$$
$$\lambda_{\boldsymbol{\theta}}(\boldsymbol{r}) := \alpha\eta(g_{\boldsymbol{\theta}}(\boldsymbol{r})). \tag{21}$$

where $\eta$ is the sigmoid function, and $\alpha$ is a scaling factor to control the range of $\lambda_{\boldsymbol{\theta}}$. $g_{\boldsymbol{\theta}} : \mathbb{R}^3 \to \mathbb{R}$ can be arbitrarily complex neural network parametrized by $\boldsymbol{\theta}$. This method has two benefits. First, by introducing additional parameters, the orbitals are more expressive and can potentially achieve better ground state energy. Second, the conventional gaussian basis function struggles to tackle the *cusps* at the location of the nuclei. By introducing the local scaling transformation, we have a scaling function that can control the sharpness of the wave functions near the nuclei, and as a consequence, this approximator can model the ground state wave function better. We perform experiments using neural local transformed orbitals on atoms. Details and results will be presented in Section 6.3.

## 5 RELATED WORK

**Deep Learning for DFT** There are several lines of work using deep learning for DFT. Some of them use neural networks in a supervised manner, i.e., use data from simulations or experiments to

fit neural networks. A representative work is conducted by (Gilmer et al., 2017), which proposes a message-passing neural network and predicts DFT results on the QM9 dataset within chemical accuracy. Custódio et al. (2019) and Ellis et al. (2021) also follow similar perspectives and predict the DFT ground state energy using feed-forward neural networks. Another active research line falls into learning the functionals, i.e. the kinetic and exchange-correlation functionals. As it is difficult to evaluate the kinetic energy given the electron density, some researchers turn to deep learning to fit a neural kinetic functional (Alghadeer et al., 2021; Ghasemi & Kühne, 2021; Ellis et al., 2021), which makes energy functional orbital-free (depending only on electron density). Achieve chemical accuracy is one of the biggest obstacles in DFT, many works (Kirkpatrick et al., 2021; Kalita et al., 2021; Ryabov et al., 2020; Kasim & Vinko, 2021; Dick & Fernandez-Serra, 2021b; Nagai et al., 2020) research into approximating the exchange-correlation functional with neural networks in the past few years, for making the DFT more accurate. It is worth noting that, the above deep learning approaches are mostly data-driven, and their generalization capability is to be tested.

**Differentiable DFT** As deep learning approaches have been applied to classical DFT methods, many researchers (Tamayo-Mendoza et al., 2018; Li et al., 2021; Kasim & Vinko, 2021; Dick & Fernandez-Serra, 2021a; Laestadius et al., 2019) have discussed how to make the training of these neural-based DFT methods, particularly the SCF loop, differentiable, so that the optimization of these methods can be easily tackled by the auto-differentiation frameworks. Li et al. (2021) uses backward mode differentiation to backpropagate through the SCF loop. In contrast, Tamayo-Mendoza et al. (2018) employs the forward mode differentiation, which has less burden on system memory compared to backward mode. Kasim & Vinko (2021) uses the implicit gradient method, which has the smallest memory footprint, however it requires running the SCF loop to convergence.

**Direct Optimization in DFT** Albeit SCF method dominates the optimization in DFT, there are researches exploring the direct optimization methods (Gillan, 1989; Van Voorhis & Head-Gordon, 2002; Ismail-Beigi & Arias, 2000; VandeVondele & Hutter, 2003; Weber et al., 2008; Ivanov et al., 2021). The challenging part is how to preserve the orthonormality constraint of the wave functions. A straightforward way to achieve this is via explicit orthonormalizaiton of the orbitals after each update (Gillan, 1989). In recent years, some have investigated direct optimization of the total energy with the orthonormality constraint incorporated in the formulation of wave functions. A representative method is to express the coefficients of basis functions as an exponential transformation of a skew-Hermitian matrix (Ismail-Beigi & Arias, 2000; Ivanov et al., 2021).

# 6 EXPERIMENTS

In this section, we demonstrate the accuracy and scalability of D4FT via numerical experiments on molecules. We compare our method with two benchmarks,

- PySCF (Sun et al., 2018): one of the most widely-used open-sourced quantum chemistry computation frameworks. It uses python/c implementation.
- JAX-SCF: our implementation of classical SCF method with Fock matrix momentum mechanism.

We implement our D4FT method with the deep learning framework JAX (Bradbury et al., 2018). For a fair comparison with the same software/hardware environment, we reimplemented the SCF method in JAX. All the experiments with JAX implementation (D4FT, JAX-SCF) are conducted on an NVIDIA A100 GPU with 40GB memory. As a reference, we also test with PySCF on a 64-core Intel Xeon CPU@2.10GHz with 128GB memory.

## 6.1 ACCURACY

We first evaluate the accuracy of D4FT on two tasks. We first compare the ground state energy obtained from D4FT with PySCF on a series of molecules. We then predict the paramagnetism of molecules to test if our method handles spin correctly. For both tasks, we use the 6-31g basis set with the LDA exchange-correlation functional.

**Ground State Energy** The predicted ground state energies are presented in Table 1. It can be seen that the absolute error between D4FT and PySCF is smaller than 0.02Ha ($\approx$0.54eV) over all the molecules compared. This validates the equivalence of our SGD optimization and SCF loop.

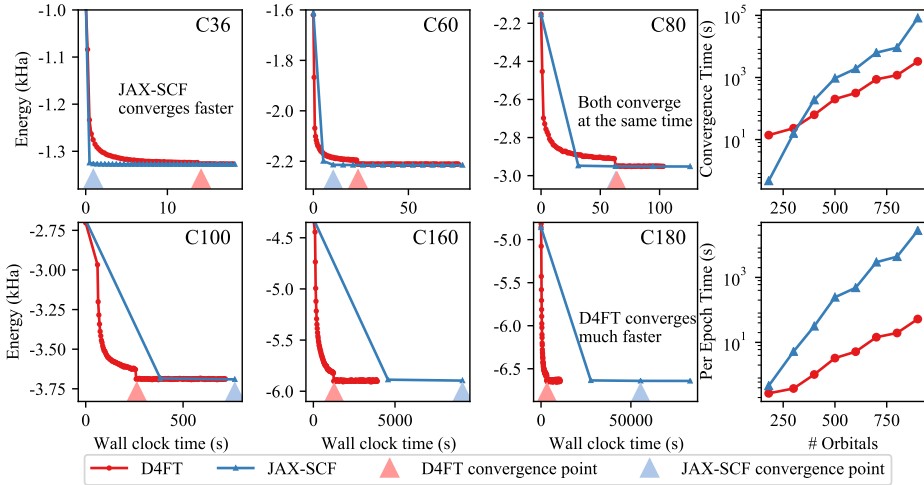

Figure 2: Convergence speed on different carbon Fullerene molecules. Columns 1-3: the convergence curves on different molecules. X-axis: wall clock time (s), Y-axis: total energy (1k Ha). Right-top: the scale of convergence time. Right-bottom: the scale of per-epoch time. It can be seen that as the number of orbitals increases, D4FT scales much better than JAX-SCF.

Table 1: Partial results of ground state energy calculation (LDA, 6-31g, Ha)

| Molecule | Hydrogen | Methane | Water | Oxygen | Ethanol | Benzene |
|---|---|---|---|---|---|---|
| PySCF | -1.03864 | -39.49700 | -75.15381 | -148.02180 | -152.01379 | -227.35910 |
| JAX-SCF | -1.03919 | -39.48745 | -75.15124 | -148.03399 | -152.01460 | -227.36755 |
| D4FT | -1.03982 | -39.48155 | -75.13754 | -148.02304 | -152.02893 | -227.34860 |

**Magnetism Prediction** We predict whether a molecule is paramagnetic or diamagnetic with D4FT. Paramagnetism is due to the presence of unpaired electrons in the molecule. Therefore, we can calculate the ground state energy with different spin magnitude and see if ground state energy is lower with unpaired electrons. We present the experimental results in Table 2. We can observe that an oxygen molecule with 2 unpaired electrons achieves lower energy, whereas carbon dioxide is the opposite, in both methods. This coincides with the fact that oxygen molecule is paramagnetic, while carbon dioxide is diamagnetic.

Table 2: Prediction on magnetism (LSDA, 6-31g, Ha)

| | Method | No unpaired electron | 2 unpaired electrons | | Method | No unpaired electron | 2 unpaired electrons |
|---|---|---|---|---|---|---|---|
| $O_2$ | PySCF | -148.02180 | **-148.09143** | $CO_2$ | PySCF | **-185.57994** | -185.29154 |
| | D4FT | -148.04391 | **-148.08367** | | D4FT | **-185.58268** | -185.26463 |

## 6.2 SCALABILITY

In this section, we evaluate the scalability of D4FT. The experiments are conducted on carbon Fullerene molecules containing ranging from 20 to 180 carbon atoms. For D4FT, we use the Adam optimizer (Kingma & Ba, 2014) with a piecewise constant learning rate decay. The initial learning rate is 0.1, it is reduced to 0.01 to 0.0001 [3] after 60 epochs. In total, we run 200 epochs for each molecule. The results are shown in Fig. 2. When the system size is small, the acceleration brought by SGD is limited. SCF has a clear advantage as the closed-form eigendecomposition brings the solution very close to optimal in just one step. Both methods become on par when the system reaches a size of 480 orbitals (C80). After that, SGD is significantly more efficient than SCF because exact

---

[3]Larger molecule gets a larger learning rate decay.

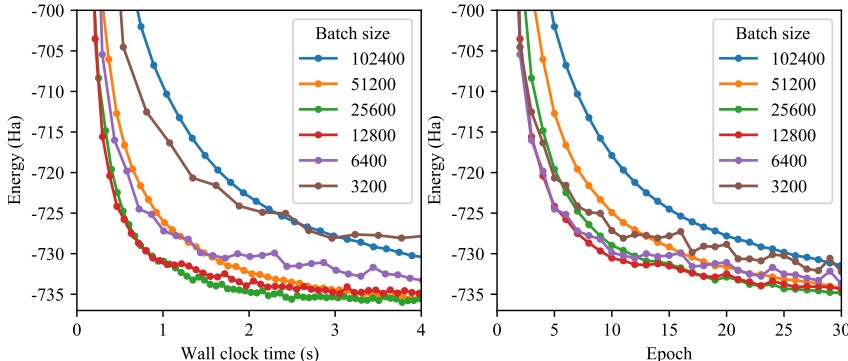

Figure 3: Convergence speed at different batch sizes on Fullerene C20. Left: Comparison under wallclock time. Right: Comparison under the number of epochs. Smaller batch sizes are less favorable in wallclock time because they can not fully utilize the GPU computation.

numerical integration is prohibitively expensive for large molecules. Column 4 in Fig. 2 shows a clear advantage of D4FT to JAX-SCF, which verifies the smaller complexity of D4FT.

**Influence of Batch Size** As repeatedly proved in large-scale machine learning, SGD with minibatch has a more favorable convergence speed as compared to full batch GD. Fig. 3 provides evidence that this is also true for D4FT. Smaller batch size generally leads to faster convergence in terms of epochs over the grids. However, when the batch size gets too small to fully utilize the GPU's computation power, it becomes less favorable in wall clock time.

## 6.3 Neural Basis

In this part, we test the neural basis with local scaling transformation presented in Sec. 4 on atoms. In this experiment, we use a simple STO-3g basis set. A 3-layer MLP with tanh activation is adopted for $g_\theta$. The hidden dimension at each layer is 9. The experimental results (partial) are shown in Table 3. Complete results are in Appendix E.3. The results demonstrate that the local scaling transformation effectively increases its flexibility of STO-3g basis set and achieves lower ground-state energy.

Table 3: Partial results of the neural basis method (LSDA, Ha)

| Atom | He | Li | Be | C | N | O |
|---|---|---|---|---|---|---|
| STO-3g | -2.65731 | -7.04847 | -13.97781 | -36.47026 | -52.82486 | -72.78665 |
| STO-3g + local scaling | **-2.65978** | **-7.13649** | **-13.99859** | **-36.52543** | **-52.86340** | **-72.95256** |

## 7 Discussion and Conclusion

We demonstrate in this paper that KS-DFT can be solved in a more deep learning native manner. Our method brings many benefits including a unified optimization algorithm, a stronger convergence guarantee, and better scalability. It also enables us to design neural-based wave functions that have better approximation capability. However, at this point, there are still several issues that await solutions in order to move further along this path. One such issue attributes to the inherent stochasticity of SGD. The convergence of our algorithm is affected by many factors, such as the choice of batch sizes and optimizers. Another issue is about the integral which involves fully neural-based wave functions. Existing numerical integration methods rely on truncated series expansions, e.g., quadrature, and Fourier transform. They could face a larger truncation error when more potent function approximators such as neural networks are used. Neural networks might not be helpful when we still rely on linear series expansions for integration. Monte Carlo methods could be what we should resort to for this problem and potentially it could link to Quantum Monte Carlo methods. On the bright side, these limitations are at the same time opportunities for machine learning researchers to contribute to this extremely important problem, whose breakthrough could benefit tremendous applications in material science, drug discovery, etc.

## ACKNOWLEDGEMENT

We would like to extend our sincerest appreciation to all those who made contributions to this research project. We express our gratitude to Pengru Huang of the NUS Institute for Functional Intelligent Materials, Alexandra Carvalho and Keian Noori from the NUS Centre for Advanced 2D Materials, Martin-Isbjorn Trappe of the NUS Center for Quantum Technologies, and Aleksandr Rodin from Yale-NUS College for providing invaluable insights and feedback during the group discussions. Their willingness to share their experiences and perspectives enabled us to gain a deeper understanding of the current computational challenges related to the quantum many-body problem at hand. We thank our research engineer, Zekun Shi, for his assistance in improving the quality of our code, paving the way towards a high-quality package. We thank Jianhao Li from University of Minnesota Goodpaster Research Group for his valuable comments on the manuscript.

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

# Appendices

## A NOTATIONS

### A.1 BRA-KET NOTATION

The bra-ket notation, or the Direc notation, is the conventional notation in quantum mechanics for denoting quantum states. It can be viewed as a way to denote the linear operation in Hilbert space. A ket $|\psi\rangle$ represent a quantum state in $\mathbb{V}$, whereas a bra $\langle\psi|$ represents a linear mapping: $\mathbb{V} \to \mathbb{C}$. The inner product of two quantum states, represented by wave functions $\psi_i$ and $\psi_j$, can be rewritten in the bra-ket notation as

$$\langle\psi_i|\psi_j\rangle := \int_{\mathbb{R}^3} \psi_i^*(\boldsymbol{r})\psi_j(\boldsymbol{r})d\boldsymbol{r}. \tag{22}$$

Let $\hat{O}$ be an operator, the expectation of $\hat{O}$ is defined as,

$$\langle\psi|\hat{O}|\psi\rangle := \int_{\mathbb{R}^3} \psi^*(\boldsymbol{r})\big[\hat{O}\psi(\boldsymbol{r})\big]d\boldsymbol{r}. \tag{23}$$

### A.2 NOTATION TABLE

We list frequently-used notations used in this paper in the next Table 4.

Table 4: Notation used in this paper

| Notation | Meaning |
|---|---|
| $\mathbb{R}$ | real space |
| $\mathbb{C}$ | complex space |
| $\boldsymbol{r}$ | a coordinate in $\mathbb{R}^3$ |
| $\psi$ | single-particle wave function/molecular orbitals |
| $\phi$ | basis function/atomic orbitals |
| $\boldsymbol{\Psi}, \boldsymbol{\Phi}$ | vectorized wave function: $\mathbb{R}^3 \to \mathbb{R}^N$ |
| $N$ | number of molecular orbitals |
| $B$ | number of basis functions |
| $\boldsymbol{x_i}, w_i$ | grid point pair (coordinate and weight) |
| $n$ | number of grid points for numerical quadrature |
| $\hat{H}$ | hamiltonian operator |
| $\rho$ | electron density |
| $\boldsymbol{B}$ | overlap matrix |
| $\sigma$ | spin index |
| $E_{\text{gs}}, E_{\text{Kin}}, \cdots$ | energy functionals |

# B KOHN-SHAM EQUATION AND SCF METHOD

## B.1 KS-DFT OBJECTIVE FUNCTION

**The Schrödinger Equation** The energy functional of KS-DFT can be derived from the quantum many-body Schrödinger Equation. The time-invariant Schrödinger Equation writes

$$\hat{H}|\Psi\rangle = \varepsilon|\Psi\rangle, \tag{24}$$

where $\hat{H}$ is the system hamiltonian and $\Psi(\boldsymbol{r})$ denotes the many-body wave function mapping $\mathbb{R}^{N\times3} \to \mathbb{C}$. $\boldsymbol{r}$ is a vector of $N$ 3D particles. The Hamiltonian is

$$\hat{H} = \underbrace{-\frac{1}{2}\nabla^2}_{\text{kinetic}} - \underbrace{\sum_i\sum_j\frac{1}{|\boldsymbol{r}_i - \boldsymbol{R}_j|}}_{\text{external}} + \underbrace{\sum_{i<j}\frac{1}{|\boldsymbol{r}_i - \boldsymbol{r}_j|}}_{\text{electron repulsion}}, \tag{25}$$

where $\boldsymbol{r}_i$ is the coordinate of the $i$-th electron and $R_j$ is the coordinate of the $j$-th nuclei. The following objective function is optimized to obtain the ground-state energy and corresponding ground-state wave function:

$$\min_{\Psi}\langle\Psi|\hat{H}|\Psi\rangle, \tag{26}$$

$$\text{s.t. } \Psi(\boldsymbol{r}) = \sigma(\mathrm{P})\Psi(\mathrm{P}\boldsymbol{r}); \ \forall\,\mathrm{P}. \tag{27}$$

**Antisymmetric Constraint** Notice there is a constraint in the above objective function, P is any permutation matrix, and $\sigma(\mathrm{P})$ denotes the parity of the permutation. This constraint comes from the Pauli exclusion principle, which enforces the antisymmetry of the wave function $Psi$. When solving the above optimization problem, we need a variational ansatz, in machine learning terms, a function approximator for $\Psi$. To satisfy the antisymmetry constraint, we can start with any function approximator $\hat{\Psi}$, and apply the antisymmetrizer $\Psi = \mathcal{A}\hat{\Psi}$, the antisymmetrizer is defined as

$$\mathcal{A}\hat{\Psi}(\boldsymbol{r}) = \frac{1}{\sqrt{N!}}\sum_{\mathrm{P}}\sigma(\mathrm{P})\hat{\Psi}(\mathrm{P}\boldsymbol{r}). \tag{28}$$

However, there are $N!$ terms summing over P, which is prohibitively expensive. Therefore, the Slater determinant is resorted to as a much cheaper approximation.

**Slater Determinant & Hartree Fock Approximation** To efficiently compute the above antisymmetrizer, the mean-field assumption is usually made, i.e. the many-body wave function is made of the product of single-particle wave functions:

$$\Psi_{\text{slater}} = \mathcal{A}\hat{\Psi}_{\text{slater}} = \mathcal{A}\prod_i\psi_i = \frac{1}{\sqrt{N!}}\begin{vmatrix} \psi_1(\boldsymbol{r}_1) & \psi_1(\boldsymbol{r}_2) & \cdots & \psi_1(\boldsymbol{r}_N) \\ \psi_2(\boldsymbol{r}_1) & \psi_2(\boldsymbol{r}_2) & \cdots & \psi_2(\boldsymbol{r}_N) \\ \vdots & \vdots & \ddots & \vdots \\ \psi_N(\boldsymbol{r}_1) & \psi_N(\boldsymbol{r}_2) & \cdots & \psi_N(\boldsymbol{r}_N) \end{vmatrix} \tag{29}$$

This approximation has a much more favorable computation complexity, computing the determinant only takes $O(N^3)$. However, it is more restricted in its approximation capability. Notice that the mean-field assumption discards the correlation between different wave functions. Therefore, with the Slater determinant approximation, it omits the correlation between electrons. Plugging this into 26 gives us the Hartree-Fock approximation.

**Orthogonal Constraints** While the Slater determinant is much cheaper to compute, integration in the $R^{N\times3}$ space is still complex. Introducing an orthogonal between the single-particle wave functions simplifies things greatly.

$$\langle\psi_i|\psi_j\rangle = \delta_{ij}. \tag{30}$$

Plugging this constraint and the Slater determinant into Eq. 26, and with some elementary calculus, there are the following useful conclusions. From this point on, without ambiguity, the symbol $r$ is also used to denote a vector in 3D space when it appears in the single-particle wave function $\psi_i(r)$.

First, the total density of the electron is the sum of the density contributed by each wave function. With this conclusion, the external part of the hamiltonian from the many body Schrödinger Equation connects with the external energy in DFT in Eq. 3.

$$\rho(r) \overset{\langle\psi_i|\psi_j\rangle=\delta_{ij}}{=\!=\!=} \sum_i^N |\psi_i(r)|^2. \tag{31}$$

Second, the kinetic energy of the wave function in the joint $N \times 3$ space breaks down into the summation of the kinetic energy of each single-particle wave function in 3D space, which again equals the kinetic term in Eq. 3.

$$\langle\Psi_{\text{slater}}| - \frac{1}{2}\nabla^2|\Psi_{\text{slater}}\rangle \overset{\langle\psi_i|\psi_j\rangle=\delta_{ij}}{=\!=\!=} \sum_i \langle\psi_i| - \frac{1}{2}\nabla^2|\psi_i\rangle. \tag{32}$$

Third, the electron repulsion term breaks into two terms, one corresponding to the hartree energy in Eq. 3, and the other is the exact exchange energy.

$$\langle\Psi_{\text{slater}}| \frac{1}{\sum_{i<j}|r_i - r_j|} |\Psi_{\text{slater}}\rangle \overset{\langle\psi_i|\psi_j\rangle=\delta_{ij}}{=\!=\!=} \tag{33}$$

$$\underbrace{\frac{1}{2}\sum_{i\neq j} \int\int \psi_i^*(r_1)\psi_j^*(r_2)\psi_i(r_1)\psi_j(r_2) \frac{1}{|r_1 - r_2|} dr_1 dr_2}_{E_H} \tag{34}$$

$$\underbrace{-\frac{1}{2}\sum_{i\neq j} \int\int \psi_i^*(r_1)\psi_j^*(r_1)\psi_i(r_2)\psi_j(r_2) \frac{1}{|r_1 - r_2|} dr_1 dr_2}_{E_X} \tag{35}$$

The above three conclusions link the Schrödinger Equation with the three terms in the KS-DFT objective. To finish the link, notice that when Slater determinant is introduced, we made the mean-field assumption. This causes an approximation error due to the electron correlation in the Hartree-Fock formulation we're deriving. In DFT, this correlation error, together with the exchange energy $E_X$ is approximated with a functional $E_{XC}$. The art of DFT is thus to find the best $E_{XC}$ that minimizes the approximation error.

## B.2 Derivation of the Kohn-Sham Equation

The energy minimization in the functional space formally reads

$$\min_{\psi_i} \left\{ E_{\text{gs}}(\psi_1, \ldots, \psi_N) \mid \psi_i \in H^1(\mathbb{R}^3), \langle\psi_i, \psi_j\rangle = \delta_{ij}, 1 \leqslant i, j \leqslant N \right\}, \tag{36}$$

where we can denote the electron density as

$$\rho(r) = \sum_{i=1}^N |\psi_i(r)|^2, \tag{37}$$

and have the objective function to be minimized:

$$E = -\frac{1}{2}\sum_i^N \int \psi_i^*(r)(\nabla^2\psi_i(r)) \, dr + \sum_i^N \int v_{\text{ext}}(r)|\psi_i(r)|^2 dr$$

$$+ \frac{1}{2}\int\int \frac{\left(\sum_i^N |\psi_i(r)|^2\right)\left(\sum_j^N |\psi_j(r')|^2\right)}{|r - r'|} dr dr' + \sum_i^N \int v_{\text{xc}}(r)|\psi_i(r)|^2 dr.$$

This objective dunction can be solved by Euler-Lagrangian method. The Lagrangian is expressive as,

$$\mathcal{L}(\psi_1, \cdots, \psi_N) = E(\psi_1, \ldots, \psi_N) - \sum_{i,j=1}^{N} \epsilon_{ij} \left( \int_{\mathbb{R}^3} \psi_i^* \psi_j - \delta_{ij} \right). \tag{38}$$

The first-order condition of the above optimization problem can be writen as,

$$\begin{cases} \dfrac{\delta\mathcal{L}}{\delta\psi_i} = 0, & i = 1, \cdots, N \\[2mm] \dfrac{\partial\mathcal{L}}{\partial\epsilon_{ij}} = 0, & i, j = 1, \cdots, N \end{cases} \tag{39}$$

The first variation above can be written as,

$$\frac{\delta\mathcal{L}}{\delta\psi_i} = \left( \frac{\delta E_{\text{Kin}}}{\delta\rho} + \frac{\delta E_{\text{Ext}}}{\delta\rho} + \frac{\delta E_{\text{Hartree}}}{\delta\rho} + \frac{\delta E_{\text{XC}}}{\delta\rho} \right) \frac{\delta\rho}{\delta\psi_i} - \sum_{j=1}^{N} \epsilon_{ij}\psi_i. \tag{40}$$

Next, we derive each components in the above equation. We follow the widely adopted tradition that treats $\psi_i^*$ and $\psi_i$ as different functions Parr (1980), and have

$$\frac{\delta\rho}{\delta\psi_i} = \frac{\delta(\psi_i^*\psi_i)}{\delta\psi_i^*} = \psi_i, \tag{41}$$

For $E_{\text{Hartree}}$, according to definition of functional derivatives:

$$\begin{aligned} E_{\text{Hartree}}[\rho + \delta\rho] - E_{\text{Hartree}}[\rho] &= \frac{1}{2} \int \int \frac{(\rho + \delta\rho)(\boldsymbol{r})(\rho + \delta\rho)(\boldsymbol{r}') - \rho(\boldsymbol{r})\rho(\boldsymbol{r}')}{|\boldsymbol{r} - \boldsymbol{r}'|} d\boldsymbol{r}d\boldsymbol{r}' \\ &= \frac{1}{2} \int \int \frac{(\delta\rho)(\boldsymbol{r})\rho(\boldsymbol{r}') + (\delta\rho)(\boldsymbol{r}')\rho(\boldsymbol{r})}{|\boldsymbol{r} - \boldsymbol{r}'|} d\boldsymbol{r}d\boldsymbol{r}' \\ &= \frac{1}{2} \cdot 2 \int \int \frac{(\delta\rho)(\boldsymbol{r})\rho(\boldsymbol{r}')}{|\boldsymbol{r} - \boldsymbol{r}'|} d\boldsymbol{r}d\boldsymbol{r}' \\ &= \int \int \frac{(\delta\rho)(\boldsymbol{r})\rho(\boldsymbol{r}')}{|\boldsymbol{r} - \boldsymbol{r}'|} d\boldsymbol{r}d\boldsymbol{r}' \\ &= \int (\delta\rho)(\boldsymbol{r}) \left( \int \frac{\rho(\boldsymbol{r}')}{|\boldsymbol{r} - \boldsymbol{r}'|} d\boldsymbol{r}' \right) d\boldsymbol{r}, \end{aligned} \tag{42}$$

where we have used the fact that:

$$\int \int \frac{f_1(\boldsymbol{r})f_2(\boldsymbol{r}')}{|\boldsymbol{r} - \boldsymbol{r}'|} d\boldsymbol{r}d\boldsymbol{r}' = \int \int \frac{f_1(\boldsymbol{r}')f_2(\boldsymbol{r})}{|\boldsymbol{r} - \boldsymbol{r}'|} d\boldsymbol{r}d\boldsymbol{r}', \tag{43}$$

due to the symmetry between $\boldsymbol{r}$ and $\boldsymbol{r}'$. We emphasize that the functional derivatives are computed on functionals mapping functions to scalars, and consequently, we need to compute that for the double integral on both $\boldsymbol{r}$ and $\boldsymbol{r}'$.

Therefore, the functional derivative for $E_{\text{Hartree}}$ can be rewritten into:

$$\frac{\delta E_{\text{Hartree}}[\rho]}{\delta\rho} = \int \frac{\rho(\boldsymbol{r}')}{|\boldsymbol{r} - \boldsymbol{r}'|} d\boldsymbol{r}'. \tag{44}$$

Using the chain rule, we have

$$\begin{aligned} \frac{\delta E_{\text{Hartree}}[\rho]}{\delta\psi^*} &= \left( \int \frac{\rho(\boldsymbol{r}')}{|\boldsymbol{r} - \boldsymbol{r}'|} d\boldsymbol{r}' \right) \frac{\partial\rho}{\partial\psi_i} \\ &= \left( \int \frac{n(\boldsymbol{r}')}{|\boldsymbol{r} - \boldsymbol{r}'|} d\boldsymbol{r}' \right) \psi_i. \end{aligned} \tag{45}$$

For $E_{\text{Ext}}$ and $E_{\text{XC}}$,

$$\begin{aligned} \frac{\delta E_{\text{Ext}}}{\delta\psi_i} &= \frac{\delta E_{\text{Ext}}}{\delta\rho} \frac{\partial\rho}{\partial\psi_i} = v_{\text{ext}}\psi_i. \\ \frac{\delta E_{\text{XC}}}{\delta\psi_i} &= \frac{\delta E_{\text{XC}}}{\delta\rho} \frac{\partial\rho}{\partial\psi_i} = v_{\text{xc}}\psi_i. \end{aligned} \tag{46}$$

For the kinetic energy,

$$\frac{\delta E_{\text{Kin}}}{\delta \psi_i^*} = \frac{-\frac{1}{2}\psi_i^* \nabla^2 \psi_i}{\delta \psi_i^*} = -\frac{1}{2}\nabla^2 \psi_i. \tag{47}$$

Taking first-order derivatives, the Lagrangian becomes:

$$H\left(\psi_1, \ldots, \psi_N\right)\psi_i = \lambda_i \psi_i, \tag{48}$$

where $\lambda_i = \sum_{j=1}^{N} \epsilon_{ij}$, and the Hamiltonian is explicitly written as

$$\hat{H} = -\frac{1}{2}\nabla^2 + v_{\text{ext}}(\boldsymbol{r}) + \int_{\mathbb{R}^3} \frac{\rho(\boldsymbol{r}')}{|\boldsymbol{r} - \boldsymbol{r}'|}d\boldsymbol{r}' + v_{\text{xc}}(\boldsymbol{r}), \tag{49}$$

$$\rho(\boldsymbol{r}) = \sum_{\sigma} \sum_{i=1}^{N} \left|\psi_i(\boldsymbol{r})\right|^2, \tag{50}$$

which is the desired form of the Kohn-Sham Equation.

## B.3   THE SCF ALGORITHM.

The connection between SCF and direct optimization can be summarized as follows. D4FT is minimizing the energy directly where the constraints are encoded into the corresponding parameter constraints, while SCF deals with the constrained optimization by Lagrangian. They are solving the same minimization problem using different approaches, which is reflected in how they deal with the constraint in the problem.

Obviously, D4FT is doing projected gradient descent on the constraint set $\Phi$. SCF is the corresponding Lagrangian, where the Lagrangian enforces the orthogonal constraint. Another constraint on the basis function is reflected in the inner product of SCF.

The SCF algorithm is presented in Algo. 1.

---
**Algorithm 1** Self-consistent Field Optimization for Kohn-Sham Equation
---
**Input:** a set of single particle wave function $\{\psi_i\}$, convergence criteria $\varepsilon$.
**Output:** ground state energy, electron density
 1: Compute the initial electron density $\rho$ via Eq. 50;
 2: **while** True **do**
 3:     Update Hamiltonian via Eq. 49;
 4:     Solve the eigenvalue equations defined in Eq. 48 and get new $\{\psi_i\}$.
 5:     Compute electron density $\rho^{\text{new}}$ via Eq. 50
 6:     **if** $|\rho^{\text{new}} - \rho| < \varepsilon$ **then**
 7:         break
 8:     **end if**
 9: **end while**

---

## C PROOFS AND DERIVATIONS

### C.1 PROOF OF PROPOSITION 3.1

*Proof.* We first prove the following statement as a preparation: (orthogonal constraint) for any $(\psi_1, \psi_2, \cdots, \psi_N)^\top \in \mathcal{F}_{\boldsymbol{W}}$, it holds that $\langle \psi_i | \psi_j \rangle = \delta_{ij}$ for all $(i, j)$. Let $(\psi_1, \psi_2, \cdots, \psi_N)^\top \in \mathcal{F}_{\boldsymbol{W}}$. From the definition,

$$\psi_i(\boldsymbol{r}) = \boldsymbol{Q}_i \boldsymbol{D} \boldsymbol{\Phi}(\boldsymbol{r}),$$

where $\boldsymbol{Q}_i$ is the $i$-th row vector of $\boldsymbol{Q}$ and $\boldsymbol{\Phi}(\boldsymbol{r}) = (\phi_1(\boldsymbol{r}), \phi_2(\boldsymbol{r}), \cdots, \phi_N(\boldsymbol{r}))^\top$. Thus, since $\boldsymbol{U}^\top \boldsymbol{U} = \boldsymbol{I}$, we have that for any $(i, j) \in [N] \times [N]$,

$$\langle \psi_i | \psi_j \rangle = \boldsymbol{Q}_i \boldsymbol{D} \boldsymbol{B} \boldsymbol{D}^\top \boldsymbol{Q}_j^\top = \boldsymbol{Q}_i \boldsymbol{\Sigma}^{-1/2} \boldsymbol{U}^\top \boldsymbol{U} \boldsymbol{\Sigma} \boldsymbol{U}^\top \boldsymbol{U} \boldsymbol{\Sigma}^{-1/2} \boldsymbol{Q}_j^\top = \boldsymbol{Q}_i \boldsymbol{Q}_j^\top = \delta_{ij}.$$

This proves the statement that for any $(\psi_1, \psi_2, \cdots, \psi_N)^\top \in \mathcal{F}_{\boldsymbol{W}}$, it holds that $\langle \psi_i | \psi_j \rangle = \delta_{ij}$ for all $(i, j)$. This statement on the orthogonal constraint and the definitions of $\mathcal{F}_{\boldsymbol{W}}$ and $\mathcal{F}$ implies that $\mathcal{F}_{\boldsymbol{W}} \subseteq \mathcal{F}$. Thus, in the rest of the proof, we need to prove $\mathcal{F}_{\boldsymbol{W}} \supseteq \mathcal{F}$. This is equivalent to

$$\{\boldsymbol{Q}\boldsymbol{D}\boldsymbol{\Phi} : (\boldsymbol{Q}, \boldsymbol{R}) = \mathrm{QR}(\boldsymbol{W}), \boldsymbol{W} \in \mathbb{R}^{N \times N}\} \supseteq \{\boldsymbol{C}\boldsymbol{\Phi} : \boldsymbol{C} \in \mathbb{R}^{N \times N}, \boldsymbol{C}_i \boldsymbol{B} \boldsymbol{C}_j^\top = \delta_{ij}\},$$

where $\boldsymbol{C}_i$ is the $i$-th row vector of $\boldsymbol{C}$. This is implied if the following holds:

$$\{\boldsymbol{Q}\boldsymbol{D} : (\boldsymbol{Q}, \boldsymbol{R}) = \mathrm{QR}(\boldsymbol{W}), \boldsymbol{W} \in \mathbb{R}^{N \times N}\} \supseteq \{\boldsymbol{C} : \boldsymbol{C} \in \mathbb{R}^{N \times N}, \boldsymbol{C}_i \boldsymbol{B} \boldsymbol{C}_j^\top = \delta_{ij}\}. \quad (51)$$

Since $\boldsymbol{D}$ is non-singular, for any $\boldsymbol{C} \in \{\boldsymbol{C} : \boldsymbol{C} \in \mathbb{R}^{N \times N}, \boldsymbol{C}_i \boldsymbol{B} \boldsymbol{C}_j^\top = \delta_{ij}\}$, there exists $\tilde{\boldsymbol{C}} \in \mathbb{R}^{N \times N}$ such that $\boldsymbol{C} = \tilde{\boldsymbol{C}} \boldsymbol{D}$ (and thus $\boldsymbol{C}_i \boldsymbol{B} \boldsymbol{C}_j^\top = \tilde{\boldsymbol{C}} \boldsymbol{D} \boldsymbol{B} \boldsymbol{D}^\top \tilde{\boldsymbol{C}}_j = \delta_{ij}$). Similarly, for any $\tilde{\boldsymbol{C}} \boldsymbol{D} \in \{\tilde{\boldsymbol{C}} \boldsymbol{D} : \tilde{\boldsymbol{C}} \in \mathbb{R}^{N \times N}, \tilde{\boldsymbol{C}}_i \boldsymbol{D} \boldsymbol{B} \boldsymbol{D}^\top \tilde{\boldsymbol{C}}_j^\top = \delta_{ij}\}$, there exists $\boldsymbol{C} \in \mathbb{R}^{N \times N}$ such that $\boldsymbol{C} = \tilde{\boldsymbol{C}} \boldsymbol{D}$ (and thus $\boldsymbol{C}_i \boldsymbol{B} \boldsymbol{C}_j^\top = \delta_{ij}$). Therefore,

$$\{\boldsymbol{C} : \boldsymbol{C} \in \mathbb{R}^{N \times N}, \boldsymbol{C}_i \boldsymbol{B} \boldsymbol{C}_j^\top = \delta_{ij}\} = \{\boldsymbol{C}\boldsymbol{D} : \boldsymbol{C} \in \mathbb{R}^{N \times N}, \boldsymbol{C}_i \boldsymbol{D} \boldsymbol{B} \boldsymbol{D}^\top \boldsymbol{C}_j^\top = \delta_{ij}\}.$$

Here, $\boldsymbol{C}_i \boldsymbol{D} \boldsymbol{B} \boldsymbol{D}^\top \boldsymbol{C}_j = \boldsymbol{C}_i \boldsymbol{\Sigma}^{-1/2} \boldsymbol{U}^\top \boldsymbol{U} \boldsymbol{\Sigma} \boldsymbol{U}^\top \boldsymbol{U} \boldsymbol{\Sigma}^{-1/2} \boldsymbol{C}_j^\top = \boldsymbol{C}_i \boldsymbol{C}_j^\top$. Thus,

$$\{\boldsymbol{C} : \boldsymbol{C} \in \mathbb{R}^{N \times N}, \boldsymbol{C}_i \boldsymbol{B} \boldsymbol{C}_j^\top = \delta_{ij}\} = \{\boldsymbol{C}\boldsymbol{D} : \boldsymbol{C} \in \mathbb{R}^{N \times N}, \boldsymbol{C}_i \boldsymbol{C}_j^\top = \delta_{ij}\}.$$

Substituting this into Eq. 51, the desired statement is implied if the following holds:

$$\{\boldsymbol{Q}\boldsymbol{D} : (\boldsymbol{Q}, \boldsymbol{R}) = \mathrm{QR}(\boldsymbol{W}), \boldsymbol{W} \in \mathbb{R}^{N \times N}\} \supseteq \{\boldsymbol{C}\boldsymbol{D} : \boldsymbol{C} \in \mathbb{R}^{N \times N}, \boldsymbol{C}_i \boldsymbol{C}_j^\top = \delta_{ij}\},$$

which is implied by

$$\{\boldsymbol{Q} : (\boldsymbol{Q}, \boldsymbol{R}) = \mathrm{QR}(\boldsymbol{W}), \boldsymbol{W} \in \mathbb{R}^{N \times N}\} \supseteq \{\boldsymbol{C} \in \mathbb{R}^{N \times N} : \boldsymbol{C}_i \boldsymbol{C}_j^\top = \delta_{ij}\}. \quad (52)$$

Finally, we note that for any $\boldsymbol{C} \in \{\boldsymbol{C} \in \mathbb{R}^{N \times N} : \boldsymbol{C}_i \boldsymbol{C}_j^\top = \delta_{ij}\}$, there exists $\boldsymbol{W} \in \mathbb{R}^{N \times N}$ such that $\boldsymbol{Q} = \boldsymbol{C}$ where $(\boldsymbol{Q}, \boldsymbol{R}) = \mathrm{QR}(\boldsymbol{W})$. Indeed, using the Gram—Schmidt process of the QR decomposition, for any $\boldsymbol{C} \in \{\boldsymbol{C} \in \mathbb{R}^{N \times N} : \boldsymbol{C}_i \boldsymbol{C}_j^\top = \delta_{ij}\}$, setting $\boldsymbol{W} = \boldsymbol{C} \in \mathbb{R}^{N \times N}$ suffices to obtain $\boldsymbol{Q} = \boldsymbol{C}$ where $(\boldsymbol{Q}, \boldsymbol{R}) = \mathrm{QR}(\boldsymbol{W})$. This implies Eq. 52, which implies the desired statement. $\qquad\square$

### C.2 PROOF OF PROPOSITION 3.2

*Proof.* The total energy minimization problem defined in Eq. 3 is a constraint minimum problem, which can be rewritten as,

$$\min_{\boldsymbol{\Psi}} \left\{ E_{\mathrm{gs}}(\boldsymbol{\Psi}) \,\Big|\, \boldsymbol{\Psi} \in H^1(\mathbb{R}^3), \langle \boldsymbol{\Psi} | \boldsymbol{\Psi} \rangle = \boldsymbol{I} \right\}, \quad (53)$$

with $H^1$ denoted as Sobolev space, which contains all $L^2$ functions whose first order derivatives are also $L^2$ integrable. People resort to the associated Euler-Lagrange equations for solution:

$$\mathcal{L}(\boldsymbol{\Psi}) = E_{\mathrm{gs}}(\boldsymbol{\Psi}) - \sum_{i,j=1}^{N} \epsilon_{ij} \left( \langle \psi_i | \psi_j \rangle - \delta_{ij} \right).$$

Letting the first-order derivatives equal to zero, i.e., $\frac{\delta \mathcal{L}}{\delta \psi_i^*} = 0$, the Lagrangian becomes: $\hat{H}(\boldsymbol{\Psi}) \psi_i = \lambda_i \psi_i$, where $\lambda_i = \sum_{j=1}^{N} \epsilon_{ij}$, and the Hamiltonian $\hat{H}$ is the same as defined in Eq. 48. To solve this nonlinear eigenvalue/eigenfunction problem, SCF iteration in Algorithm 1 is adopted. The

output of the $(t + 1)$th iteration is derived from the Hamiltonian with the previous output, i.e., $\hat{H}\left(\boldsymbol{\Psi}^t\right)\psi_i^{t+1} = \lambda_i^{t+1}\psi_i^{t+1}$. The convergence criterion is $\boldsymbol{\Psi}^t = \boldsymbol{\Psi}^{t+1}$, which is the self-consistency formally defined in Definition 3.1.

Since GD and SCF are solving essentially the same problem, we can derive the equivalence between the solutions of GD and SCF. GD is doing projected gradient descent on the constraint set $\{\boldsymbol{\Psi} \mid \langle\boldsymbol{\Psi}|\boldsymbol{\Psi}\rangle = \boldsymbol{I}\}$. SCF is the corresponding Lagrangian, where the Lagrangian enforces the orthogonal constraint. $\qquad\square$

# D COMPLEXITY ANALYSIS

## D.1 COMPLEXITY ANALYSIS FOR FULL BATCH

SCF's computation complexity analysis

$$\int \int \rho(\boldsymbol{r}_1) \frac{1}{|\boldsymbol{r}_1 - \boldsymbol{r}_2|} \psi_i(\boldsymbol{r}_2) \psi_j(\boldsymbol{r}_2) d\boldsymbol{r}_1 d\boldsymbol{r}_2$$

1. Compute $\psi_i(\boldsymbol{r})$ for $i \in [1 .. N]$, and all the $n$ grid points. it takes $\mathcal{O}(NB)$ to compute for each $r$, and we do it for $n$ points. Therefore, it takes $\mathcal{O}(nNB)$, producing a $\mathbb{R}^{n \times N}$ matrix.

2. Compute $\rho(\boldsymbol{r})$, it takes an extra $\mathcal{O}(nN)$ step to sum the square of the wave functions, on top of step 1, produces $\mathbb{R}^n$ vector.

3. Compute $\psi_i(\boldsymbol{r})\psi_j(\boldsymbol{r})$, it takes an outer product on top of step 1, $\mathcal{O}(nN^2)$, produces $\mathbb{R}^{n \times N \times N}$ tensor.

4. Compute $\frac{1}{|\boldsymbol{r}_1 - \boldsymbol{r}_2|}$, $\mathcal{O}(n^2)$.

5. Tensor contraction on outcomes from step 2, 3, 4. $\mathcal{O}(n^2 N^2)$.

6. In total $\mathcal{O}(nNB) + \mathcal{O}(n^2 N^2) = \mathcal{O}(n^2 N^2)$.

SGD's computation complexity analysis

$$\int \int \rho(\boldsymbol{r}_1) \frac{1}{|\boldsymbol{r}_1 - \boldsymbol{r}_2|} \rho(\boldsymbol{r}_2) d\boldsymbol{r}_1 d\boldsymbol{r}_2$$

1. Using steps 1, 2, and 4 from SCF. Step 3 is not used because only energy is needed, $\mathcal{O}(nNB)$.

2. Contract $\mathbb{R}^n$, $\mathbb{R}^n$, $\mathbb{R}^{n \times n}$ into a scalar, $\mathcal{O}(n^2)$.

3. In total $\mathcal{O}(nNB) + \mathcal{O}(n^2) = \mathcal{O}(nN^2) + \mathcal{O}(n^2)$.

## D.2 COMPLEXITY ANALYSIS FOR MINIBATCH

The complexity of SCF under minibatch size $m$ is $\mathcal{O}(mNB) + \mathcal{O}(m^2 N^2)$, since the number of iteration per epoch is $n/m$, the total complexity for one epoch is $\mathcal{O}(nNB) + \mathcal{O}(nmN^2) = \mathcal{O}(n(1 + m)N^2)$ since $B = N$, i.e., the number of orbitals equal to that of the basis.

Similarly, the complexity of GD is $(\mathcal{O}(mNB) + \mathcal{O}(m^2))n/m = \mathcal{O}(n(N^2 + m))$.

## D.3 LITERATURE ON THE CONVERGENCE OF SCF METHODS.

There is a rich literature on the convergence of SCF. Cancès et al. (2021) provides a literature review and method summarization for the iterative SCF algorithm. Specifically, they show that the problem can be tackled via either Lagrangian + SCF or direct gradient descent on matrix manifold. Yang et al. (2007) shows that the Lagrangian + SCF algorithm is indirectly optimizing the original energy by minimizing a sequence of quadratic surrogate functions. Specifically, at the $t$th time step, SCF is minimizing the quadrature surrogate energy using the $t$th Hamiltonian, $\min_A \frac{1}{2} \text{Tr}(A^* H^{(t)} A)$, while the correct nonlinear energy should be $\min_A \frac{1}{2} \text{Tr}(A^* H(A) A)$. So, our GD directly minimizes the nonlinear energy, while SCF deals with a quadrature surrogate energy. Using a concrete two-dimensional example, how SCF may fail to converge is illustrated, since the second order approximation of the original. Yang et al. (2009) shows that SCF will converge to two different limit points, but neither of them is the solution to a particular class of nonlinear eigenvector problems. Nevertheless, they also identify the condition under which the SCF iteration becomes contractive which guarantees its convergence, which is the gap between the occupied states and unoccupied states is sufficiently large and the second order derivatives of the exchange-correlation functional are uniformly upper bounded. Bai et al. (2020) provides optimal convergence rate for a class of nonlinear eigenvector directly minimizesly; since SCF is an iterative algorithm, we can show that

SCF is a contraction mapping:

$$\limsup_{t \to \infty} \frac{d(A^{(t+1)}, A^*)}{d(A^{(t)}, A^*)} \leq \eta, \tag{54}$$

where $d$ measures the distance between two matrices, $A^*$ is the solution, and $\eta$ is the contraction factor determining the convergence rate of SCF, whose upper bound is given in their result. Liu et al. (2015; 2014) formulates SCF as a fixed point map, whose Jacobian is derived explicitly, from which the convergence of SCF is established, under a similar condition to the one in Yang et al. (2009). Cai et al. (2018) derive nearly optimal local and global convergence rate of SCF.

# E  MORE EXPERIMENTAL RESULTS

## E.1  FULL VERSION OF TABLE 1

Table 5: Comparison of Ground State Energy (LDA, 6-31g, Ha)

| Molecule | Method | Nuclear Repulsion Energy | Kinetic+ External Energy | Hartree Energy | XC Energy | Total Energy |
|---|---|---|---|---|---|---|
| Hydrogen | PySCF | 0.71375 | -2.48017 | 1.28032 | -0.55256 | -1.03864 |
| | D4FT | 0.71375 | -2.48343 | 1.28357 | -0.55371 | -1.03982 |
| | JaxSCF | 0.71375 | -2.48448 | 1.28527 | -0.55390 | -1.03919 |
| Methane | PySCF | 13.47203 | -79.79071 | 32.68441 | -5.86273 | -39.49700 |
| | D4FT | 13.47203 | -79.77446 | 32.68037 | -5.85949 | -39.48155 |
| | JaxSCF | 13.47203 | -79.80889 | 32.71502 | -5.86561 | -39.48745 |
| Water | PySCF | 9.18953 | -122.83945 | 46.58941 | -8.09331 | -75.15381 |
| | D4FT | 9.18953 | -122.78656 | 46.54537 | -8.08588 | -75.13754 |
| | JaxSCF | 9.18953 | -122.78241 | 46.53507 | -8.09343 | -75.15124 |
| Oxygen | PySCF | 28.04748 | -261.19971 | 99.92152 | -14.79109 | -148.02180 |
| | D4FT | 28.04748 | -261.13046 | 99.82705 | -14.77807 | -148.03399 |
| | JaxSCF | 28.04748 | -261.16314 | 99.87551 | -14.78290 | -148.02304 |
| Ethanol | PySCF | 82.01074 | -371.73603 | 156.36891 | -18.65741 | -152.01379 |
| | D4FT | 82.01074 | -371.73431 | 156.34982 | -18.65517 | -152.02893 |
| | JaxSCF | 82.01074 | -371.64258 | 156.27433 | -18.65711 | -152.01460 |
| Benzene | PySCF | 203.22654 | -713.15807 | 312.42026 | -29.84784 | -227.35910 |
| | D4FT | 203.22654 | -712.71081 | 311.94665 | -29.81097 | -227.34860 |
| | JaxSCF | 203.22654 | -712.91053 | 312.15690 | -29.84042 | -227.36755 |

## E.2  FULL VERSION OF TABLE 3

Table 6: Comparison of ground state energy on atoms (LSDA, STO-3g, Hartree).

| Atom | He | Li | Be | B | C |
|---|---|---|---|---|---|
| PySCF | -2.65731 | -7.06641 | -13.98871 | -23.66157 | -36.47199 |
| D4FT | -2.65731 | -7.04847 | -13.97781 | -23.64988 | -36.47026 |
| NBLST | **-2.65978** | **-7.13649** | **-13.99859** | **-23.67838** | **-36.52543** |

| Atom | N | O | F | Ne | Na |
|---|---|---|---|---|---|
| PySCF | -52.80242 | -72.76501 | -96.93644 | -125.38990 | -158.40032 |
| D4FT | -52.82486 | -72.78665 | -96.99325 | -125.16031 | -158.16140 |
| NBLST | **-52.86340** | **-72.95256** | **-97.24058** | **-125.62643** | -157.72314 |

| Atom | Mg | Al | Si | P | S |
|---|---|---|---|---|---|
| PySCF | -195.58268 | -237.25091 | -283.59821 | -334.79401 | -390.90888 |
| D4FT | -195.47165 | -237.03756 | -283.41116 | -335.10473 | -391.09967 |
| NBLST | **-196.00629** | -235.83420 | **-283.72964** | **-335.38168** | **-391.34567** |

## E.3  MORE RESULTS ON THE EFFICIENCY COMPARISON

We test the efficiency and scalability of our methods against a couple of mature Quantum Chemistry softwares. We compute the ground-state energy using each implementation and record the wall-clock running time. The softwares we compare with are:

- PySCF. PySCF is implemented mainly with C (Qint and libxc), and use python as the interface.
- GPAW. A Plane-wave method for solid and molecules.
- Psi4. Psi4 is mainly written in C++.

We use an sto-3g basis set and a lda exchange-correlation functional. For GPAW, we apply the lcao mode, and use dzp basis set. The number of bands calculated is the double of the number of electrons. The number of grids for evaluating the XC and Coulomb potentials is 400*400*400 by default. The maximum number of iteration is 100 for all the methods. Our D4FT ran on an NVIDIA A100 40G GPU, whereas other methods did on an Intel Xeon 64-core CPU with 60G memory. The wall-clock running time for convergence (by default setting) are shown in the following table.

Table 7: Comparison of wall-clock running time (s).

| Molecule | C20 | C60 | C80 | C100 | C180 |
|---|---|---|---|---|---|
| D4FT | 11.46 | 23.14 | 62.98 | 195.13 | 1103.19 |
| PySCF | 20.75 | 186.67 | >3228.20 | 672.98 | 1925.25 |
| GPAW | >2783.08 | – | – | – | – |
| Psi4 | >46.12 | 510.35 | >2144.49 | 2555.16 | 14321.64 |

Notation $>$ represent that the experiment cannot finish or converge given the default convergence condition. It can be seen from the results that our D4FT is the fastest among all the packages.

### E.4 CASES THAT PYSCF FAILS.

We show two cases that do not converge with PySCF. We test two unstable systems, C120 and C140, which do not exist in the real world. Both of them are truncated from a C180 Fullerene molecule. Results are shown in Fig. 4.

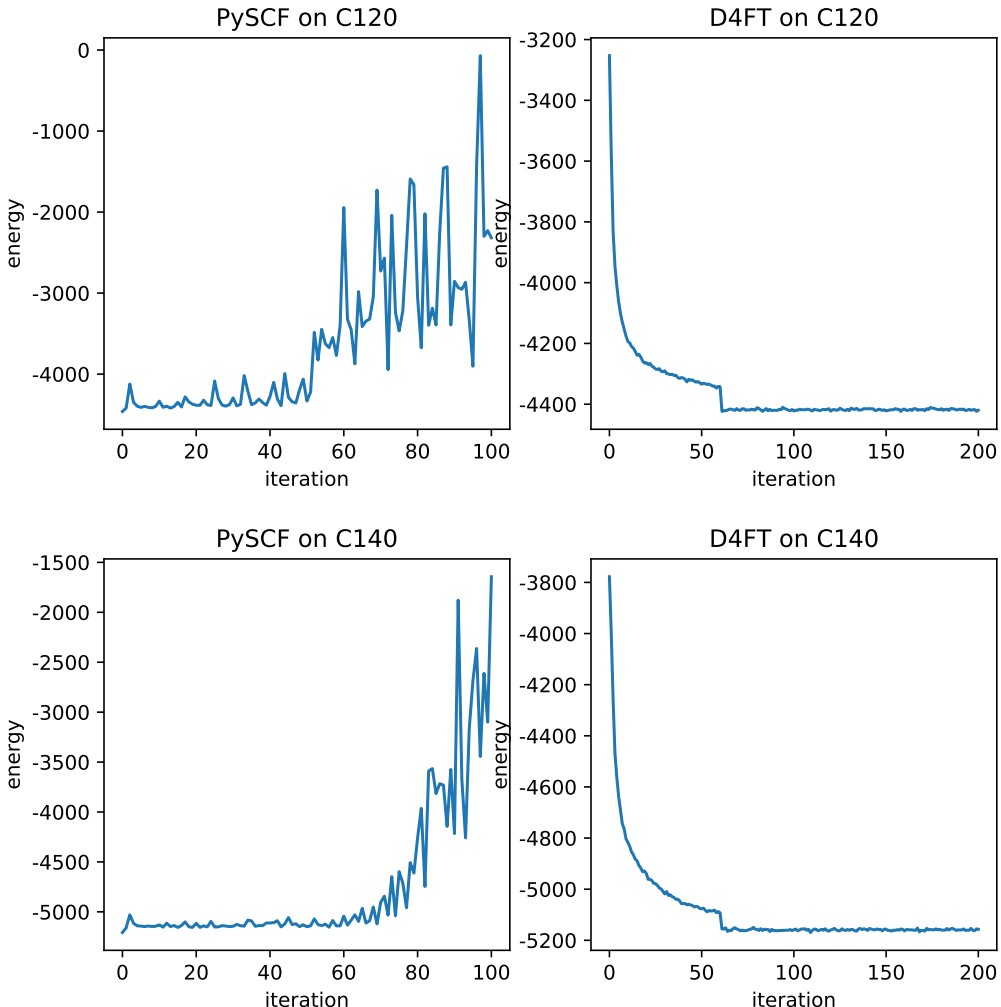

Figure 4: Cases that do not converge with PySCF, but do with D4FT. Due to the good convergence property of SGD, D4FT is more likely to converge.

