# OpenReview forum: "D4FT: A Deep Learning Approach to Kohn-Sham Density Functional Theory"
_ICLR.cc/2023/Conference — ICLR 2023 notable top 25%_

### Official Review · Reviewer_GibD · 2022-10-23

**Confidence:** 1
**Correctness:** 4
**Technical Novelty And Significance:** 2
**Empirical Novelty And Significance:** 2
**Recommendation:** 6

**Clarity, Quality, Novelty And Reproducibility:**

It looks like the paper is well organized but I have no idea on the novelty and reproducibility.

**Strength And Weaknesses:**

Unfortunately, I am not in this area so cannot identify the possible strength and weaknesses.

**Summary Of The Paper:**

This paper proposes a deep learning model for Kohn-Sham density functional theory. Unfortunately, I know nothing about this theory and related works, so I cannot accurately summaries this work.

**Summary Of The Review:**

Unfortunately, I am not in this area.

---

> ### Author Response · Authors · 2022-11-10
> **Thank you for reviewing our paper.**
>
> Thank you for the time and effort putting into reviewing this paper, and we appreciate your honest assessment.  We would like to briefly introduce the background of DFT. The Kohn-Sham DFT is a Nobel prize-winning method that is widely used in quantum chemistry and computational material science. This method fosters a lot of scientific discovery including finding catalysts, drug discovery, and Li-ion battery design. Despite its great value in scientific research, the current computational efficiency hinders its applications in many large-scale systems, such as superconductive Graphene material design, and large biological molecular structure prediction, which often takes days or months to converge. In this work, we explore the idea of applying deep learning techniques to solve this essential scientific problem. Our method brings many benefits including a unified optimization algorithm, a stronger convergence
> guarantee, and better scalability.

---

### Official Review · Reviewer_BHRj · 2022-10-23

**Confidence:** 2
**Correctness:** 4
**Technical Novelty And Significance:** 3
**Empirical Novelty And Significance:** 3
**Recommendation:** 5

**Clarity, Quality, Novelty And Reproducibility:**

The paper is clearly written. Even readers unfamiliar with quantum physics can understand most parts of the paper. The modeling for the basis functions is an interesting contribution. However, the theoretical novelty is limited. The algorithm proposed in the paper is simple. However, the reproducibility is unclear as the authors did not share the codes they used in their experiments.

**Strength And Weaknesses:**

Strength
- Overall the paper is clearly written. The significance of the problem is well introduced though the problem setting may not be well-known in the machine learning community. The modeling of the local scaling transformation using NNs is interesting.

 Weaknesses
- There are some unclear points in the algorithm. The Jacobian appears in (19), which depends on a NN model. It is unclear how to compute the Jacobian matrix efficiently. I guess the back-propagation for NNs is applied to calculate the derivative w.r.t. the input vector instead of the weight parameters. A detailed description of the algorithm would be helpful to readers.
- The authors provide theoretical contributions as some propositions, but the theoretical results are relatively straightforward.
- Numerical experiments are not sufficient. Hyper-parameter tuning is important to achieve high accuracy in the computation. In the discussion, the author pointed out, "One potential remedy for this problem is the second-order optimizers." However, the current paper does not provide a systematic study of that problem. Ablation studies should be provided to assess which factor is significant to attain satisfactory results.


**Summary Of The Paper:**

This paper studies the computation of wave functions in Kohn-Sham Density Functional Theory (KS-DFT). Differently from conventional works, the authors use neural network (NN) models with a stochastic optimization method in machine learning to tackle the problem. The problem is formulated as the minimization problem of energy function depending on the set of orthogonal wave functions. The integral in the energy function is approximated by the finite sum based on the numerical quadrature. The stochastic gradient descent (SGD) is then employed to approximate the optimization for the objective function consisting of a finite sum. The parameter transformation with the Jacobian is introduced to deal with the orthogonal constraint among wave functions. The neural network model is used as the model for the transformation. Numerical experiments show that the proposed method provides comparable accuracy and high scalability to exciting methods.

**Summary Of The Review:**

The paper is clearly written. The neural basis function is an interesting model to deal with the problem considered in the paper. However, the theoretical insight is limited. Showing other applications of neural basis function would be beneficial for readers.

---

> ### Author Response · Authors · 2022-11-10
> **Author's response**
>
> Thank you for your dedicated time and effort in reviewing the paper. All the comments are very helpful and we have revised our paper accordingly. Before addressing the concerns of the reviewer, we would like to stress that our main novelty lies in reformulating KS-DFT in the deep learning way, it is a major shift in algorithm compared to existing works that mainly rely on SCF. With the above change, further integration of machine-learned components can work in a fully end-to-end manner, for example, integrating neural-based xc functionals. In this paper, we provide the neural scaling orbital as an example of this benefit.
>
> We now address the concerns of the reviewer.
>
> > ***There are some unclear points in the algorithm. The Jacobian appears in (19), which depends on a NN model. It is unclear how to compute the Jacobian matrix efficiently. I guess the back-propagation for NNs is applied to calculate the derivative w.r.t. the input vector instead of the weight parameters.***
>
> Thank you for pointing this out. As the neural local scaling is transforming 3D space, the Jacobian of NN $(R^3 \to R)$ in neural local scaling can be cheaply calculated with the existing auto-differentiation frameworks in either forward mode or backward mode (jax.jacfwd and jax.jacrev for example). The input/output dimensions of NN are 3-dimensional and thus full computation of the determinant of a 3-dimensional matrix is affordable. We will make this point clear in the draft.
>
>
> > ***The authors provide theoretical contributions as some propositions, but the theoretical results are relatively straightforward.***
>
> Admittedly, we did not invent any new techniques in the theoretical analysis. However, whether direct minimization will achieve self-consistency is not straightforward. As we’re comparing with the self-consistent field method, it is necessary to answer the question of whether direct minimization is also achieving self-consistency. Studies on self-consistency are still active and it has not been fully understood [1, 2].
>
> Another difficulty of our method is that the loss function of our approach is not a simple MSE or cross entropy-type function, but a complicated solution to the many-body Schrodinger equation with HF approximation and orthogonal constraint. Theoretical analysis for such loss function is arduous.
>
> > ***The algorithm proposed in the paper is simple.***
>
> We agree that we do not invent any new techniques from the machine learning perspective. However, we do make a major methodological shift for the KS-DFT, which makes it fully differentiable and easier to integrate deep learning components. We hope the reviewer can evaluate our work from the perspective of its potential scientific applications instead of a pure machine learning methodology.
>
> >***Numerical experiments are not sufficient. Hyper-parameter tuning is important to achieve high accuracy in the computation.***
>
> Thank you for pointing this out. We apologize for this confusion, the hyper-parameter tuning we mentioned is about the learning rate annealing. The annealing schedule is currently hand-tuned as can be seen in our figure. We have rephrased this statement in the new draft. As for accuracy, KS-DFT is a pure optimization problem than a generalization problem, the solved ground state energies are very stable, and can be seen that the results align well with SCF in all cases.
>
> > ***Showing other applications of neural basis function would be beneficial for readers.***
>
> We like to reiterate that the main scope of this paper is to present a deep learning alternative to the framework that could bypass the SCF method. The neural local scaling is introduced to support our claim that this framework is more flexible for integrating more machine learning techniques. More sophisticated designs on the neural-based orbitals are not our main goal.
>
> On the other hand, we agree with the reviewer that this line of research can be further explored.
> The exploration of more expressive basis functions is one of the key research problems in quantum chemistry and computational condensed matter physics. Neural networks are actually one of the most promising options in terms of fitting power. However, eigendecomposition in conventional SCF can only solve linear coefficients, therefore we believe our framework makes it easier for future research into this problem. We would also like to note that for efficiency, the overlap matrix S shall be preserved. This constraint makes the problem non-trivial.
>
> **Reference**
> - [1] Bai, Z., Li, R. C., & Lu, D. (2020). Optimal convergence rate of self-consistent field iteration for solving eigenvector-dependent nonlinear eigenvalue problems. arXiv preprint arXiv:2009.09022.
> - [2] Kuang, Y., & Hu, G. (2020). On stabilizing and accelerating SCF using ITP in solving Kohn–Sham equation. Communications in Computational Physics, 28(3), 999-1018.
> - [3] Cancès et al. Computational quantum chemistry: A primer. 2003. Page 78-79.

---

### Official Review · Reviewer_fw91 · 2022-10-24

**Confidence:** 4
**Correctness:** 3
**Technical Novelty And Significance:** 3
**Empirical Novelty And Significance:** 3
**Recommendation:** 6

**Clarity, Quality, Novelty And Reproducibility:**

The paper is well organized and clearly written. The mathematical derivation is solid. Since several machine learned DFT schemes were developed previously, the novelty of a new one is largely determined by its efficiency. The corresponding information await replenishment. The calculations presented in this manuscript are reasonable and reproducible.

**Strength And Weaknesses:**

Strength:

The approach proposed in this paper was proven to has the same expressivity as the SCF method, while reducing the computational complexity from O(N4) to O(N3). The methodology is expected to be also suitable for more complex neural-based wave functions.

Weaknesses:

1. For a fair comparison with the same software/hardware environment, the authors reimplemented the SCF method in JAX. Nevertheless, its equivalence with the conventional SCF solvers in terms of efficiency and scalability was not clarified.

2. To demonstrate the efficiency and stability, the authors carried out some calculating experiments. However, the coverage of experiments was not sufficient. When evaluating the scalability of D4FT, carbon Fullerene molecules containing ranging from 20 to 180 carbon atoms were simulated. The size of the system seems to be not large enough. As the main limitation of the conventional SCF is its inefficiency for large molecules or solid cells, a machine learning based method is expected to behave better on large scale systems. This part of data might be implemented to show its advantage in reducing the computational complexity from O(N4) to O(N3).

3. The approach raised was claimed to bring a stronger convergence guarantee compared with conventional SCF, while this was not supported by a related instance. A successfully converged calculation done by D4FT which failed in conventional SCF would be valuable.

**Summary Of The Paper:**

In this work, a deep learning approach for solving KS-DFT was proposed, which converts the objective function for KS-DFT into an unconstrained equivalent by reparameterizing the orthogonal constraints as a feed-forward computation. By using stochastic gradient descent, the integral was amortized over the optimization steps. Its equivalence to the conventional SCF method was proven both empirically and theoretically. The comparison between this work and some other deep-learning-assisted DFT approaches were discussed. The accuracy and scalability were tested via experiments on molecules by comparing with two benchmarks (PySCF and JAX-SCF). The authors further raised several issues that await solutions based on the limitaions from this paper.

**Summary Of The Review:**

The paper developed a deep learning approach for solving KS-DFT by converting the objective function for KS-DFT into an unconstrained equivalent by reparameterizing the orthogonal constraints as a feed-forward computation. Its equivalence to the conventional SCF method is proven both empirically and theoretically. The presentation in the manuscript is clear. I recommend it for publication if authors address the above mentioned comments and offer some more substantial experimental data.

---

> ### Author Response · Authors · 2022-11-10
> **Author's response**
>
> First, we would like to thank the reviewer for the in-depth and valuable comments which helps improve the quality of this manuscript significantly. Here is the response to each question.
>
>  >***The equivalence of JAX-SCF with the conventional SCF solvers in terms of efficiency and scalability was not clarified.***
>
> Thank you for pointing this out. First of all, we may want to stress that the purpose of implementing the JAX version of SCF is for an apple-to-apple comparison of D4FT and SCF on GPU. The scope of this paper is to compare direct SGD with SCF, therefore we implemented both in the vanilla form, without further optimization. In existing DFT softwares, there are many numerical optimizations, e.g. 8-fold symmetry in two-electron integral, Direct Inversion in the Iterative Subspace (DIIS), and Rys quadrature, which significantly reduce the running time. We admit that we’ll need to take in all these factors to have a completely fair comparison. Notably, these numerical optimization methods can be potentially applied to our implementation, and we will explore these methods in our future works.
>
> To provide a reference to existing DFT implementations, we conducted experiments on solving KS-DFT Carbon Fullerene molecules against several mainstream implementations: PySCF, Psi4, and GPAW. Please note that PySCF, as well as GPAW and Psi4, are mature packages that have been optimized for years by many developers. We use an STO-3g basis set and an LDA exchange-correlation functional. D4FT is run on an NVIDIA A100 40G GPU, whereas other methods run on an Intel Xeon 64-core CPU with 64G memory. The wall-clock running time (in seconds) for convergence (by default setting) is shown in the following table.
>
> | Package | C20 | C60 | C80 | C100 | C180 |
> |-----------|------ |------ |------  |   ------|  ------|
> |D4FT    |11.46|23.14|62.98|195.13|1103.19|
> |PySCF  |20.75|186.67|>3228.20|672.98|1925.25|
> |GPAW   |>2783.08|--|--|--|--|
> |Psi4    |>46.12|510.35|>2144.49| 2555.16|14321.64|
>
> This result is included in Appendix E.3 (Page 22).
>
>
> >***Machine learning based method is expected to behave better on large scale systems.***
> >***However, the coverage of experiments was not sufficient. The size of the system seems to be not large enough.***
>
> We’d like to clarify that the DF4T method described is **ab-initio** rather than machine learning. Our method is more of a reimplementation of KS-DFT in the deep learning flavor.
>
> We agree that machine-learned models usually enjoy better scalability. There are usually two types of approaches that apply neural networks to Kohn-Sham DFT. The **data-driven** methods usually fit a neural network to make direct predictions. These methods can scale up to very large datasets [1, 2] but their generalization capability is in doubt. In contrast, our method is NOT date-driven but ab-initio, as it does not require any data but solves the quantum system from the first principle. **In this regard, we think C180 is already a reasonably sized system.** Previous works in differentiable DFT have smaller sizes, e.g. [3] computes up to fluorine, which has two F atoms, and [4] up to C6H8O6 which contains 20 atoms.
>
> >***A successfully converged calculation done by D4FT which failed in conventional SCF would be valuable.***
>
> Thank you for this very constructive suggestion. We have included two cases that do not converge with PySCF. Please kindly find the results in Appendix E.4 on Page 24. Note that on JAX-SCF we added a simple momentum trick to make them converge, in contrast, in D4FT, SGD just works out of the box.
>
> - [1] Zhang, S., Liu, Y., & Xie, L. (2020). Molecular mechanics-driven graph neural network with multiplex graph for molecular structures. arXiv preprint arXiv:2011.07457.
> - [2] Atz, K., Grisoni, F., & Schneider, G. (2021). Geometric deep learning on molecular representations. Nature Machine Intelligence, 3(12), 1023-1032.
> - [3] Kasim, M. F., & Vinko, S. M. (2021). Learning the exchange-correlation functional from nature with fully differentiable density functional theory. Physical Review Letters, 127(12), 126403.
> - [4] Kasim, M. F., Lehtola, S., & Vinko, S. M. (2022). DQC: A Python program package for differentiable quantum chemistry. The Journal of chemical physics, 156(8), 084801.

---

> > ### Comment · Reviewer_fw91 · 2022-12-10
> > **Thank you for the thorough response**
> >
> > I really appreciate your effort giving a response to my review. I have carefully read it as well as the other reviews and responses. I think this work contains some contributions that deserve to be published, so I've updated the score.

---

### Author Response · Authors · 2022-11-15
**Looking forward to further discussion**

Dear Reviewers,

Thank you again for your in-depth and valuable comments, which are very helpful for us. We have uploaded a modified version of the manuscript and have integrated our response to all reviewers. We believe that the new version addresses all the important points that were raised, but we will be happy to make additional changes to address any remaining concerns.

We totally understand that this is quite a busy period, and reviewing papers requires a lot of time and effort. As the end of the discussion is approaching, we are eager to hear more about your comments on our manuscript and our response. Thank you for engaging in the discussion so far, and please let us know if you have any final questions that we can address.

Best regards,

The authors

---

### Decision · Program_Chairs · 2023-01-20

**Decision:**

Accept: notable-top-25%

**Justification For Why Not Higher Score:**

Needs more scalability for it to be practically useful.

**Justification For Why Not Lower Score:**

Important topic. Significant results. No major pending criticism from the reviewers.

**Metareview: Summary, Strengths And Weaknesses:**

The paper proposes a new approach for solving Kohn-Sham Density Functional Theory (KS-DFT) based on deep neural nets. This is typically done via an SCF method which solves for a constrained optimization problem. Instead, the paper proposes the D4FT method solves for an end-to-end differentiable objective that minimizes the total energy of the system. In doing so, the authors (a) reparameterize the constraints to signify a feedforward computation, (b) approximate the integral via a numerical sum over a grid and approximate it via a random mini-batch. The deep learning based approach reduces the computational complexity from biquadratic to cubic without any loss in expressivity. Experiments demonstrate higher accuracy and relatively higher stability compared to past work. On the positive side, the topic is of immense significance for various scientific fields, the paper's contributions are simple, elegant, and clear. There are no major concerns, except for perhaps more ablations as desired by one of the reviewers.

**Note From Pc:**

if the above contains the word "oral" or "spotlight" please see: "oral" presentation means -> notable-top-5% and "spotlight" means -> notable-top-25%. As stated in our emails, we are disassociating presentation type from AC recommendations